

# Genome-wide analyses of the bHLH gene family reveals structural and functional characteristics in the aquatic plant *Nelumbo nucifera*

Tian-Yu Mao[1,2], Yao-Yao Liu[1,2], Huan-Huan Zhu[1,2], Jie Zhang[1,2], Ju-Xiang Yang[1,2], Qiang Fu[1,2], Nian Wang[1,2] and Ze Wang[1,2]

[1] Key Laboratory of Horticultural Plant Biology, College of Horticulture and Forestry Sciences, Huazhong Agriculture University, Wuhan, China
[2] Key Laboratory of Urban Agriculture in Central China, Ministry of Agriculture and Rural Affairs, Wuhan, China

## ABSTRACT

Lotus (*Nelumbo nucifera* Gaertn.) is an economically important aquatic plant with multiple applications, but water salinity and cold stress seriously affect lotus yield and distribution. The basic helix-loop-helix (bHLH) transcription factors (TFs) play a vital role in plant growth and development, metabolic regulation processes and responses to environmental changes. However, systematic analyses of the bHLH TF family in lotus has not yet been reported. Here, we report the identification and description of bHLH genes in lotus (*NnbHLHs*) with a focus on functional prediction, particularly for those involved in stress resistance. In all, 115 *NnbHLHs* were identified in the lotus genome and classified into 19 subfamilies. The chromosomal distribution, physicochemical properties, bHLH domain, conserved motif compositions and evolution of these 115 *NnbHLHs* were further analyzed. To better understand the functions of the lotus bHLH family, gene ontology, cis-element, and phylogenetic analyses were conducted. *NnbHLHs* were predicted to be involved in plant development, metabolic regulation and responses to stress, in accordance with previous findings. Overall, 15 *NnbHLHs* were further investigated with functional prediction via quantitative real-time PCR analyses. Meanwhile, expression profiles of *NnbHLHs* in four tissues indicated that many *NnbHLHs* showed tissue preference in their expression. This study is supposed to provide a good foundation for further research into the functions and evolution of *NnbHLHs*, and identifies candidate genes for stress resistance in lotus.

## INTRODUCTION

The basic helix-loop-helix (bHLH) family is the second largest gene family in plants, after the MYB family (*Feller et al., 2011*). With the greater availability of genome sequence data, genes in the bHLH family have been identified and characterized in various plant species (*Li et al., 2006*; *Niu et al., 2017*; *Sun, Fan & Ling, 2015*; *Toledo-Ortiz,*

Corresponding author
Jie Zhang,
flybebrave@mail.hzau.edu.cn

*Huq & Quail, 2003*). The bHLH gene family is named after its distinctive structure of a bHLH domain, consisting of two conserved regions, a basic region and a helix–loop–helix region (HLH region), of approximately 60 amino acids. The basic region is located at the N-terminus of the bHLH domain and consists of approximately 17 amino acids, six of which are basic amino acid residues. This domain is a DNA-binding region that enables bHLH transcription factors (TFs) to bind to E-box (CANNTG) (*Carretero-Paulet et al., 2010*; *Niu et al., 2017*). The HLH region, containing two amphipathic α-helices linked by a loop region with a variable sequences (*Murre, Mccaw & Baltimore, 1989*), is located at the C-terminus and participates in protein dimerization (*Brownlie et al., 1997*). The regions outside the bHLH domain are distinctly divergent. In animals, the bHLH genes are classified into six groups (from A to F), which contain 45 subgroups based on their various functions in the regulation of gene expression, target DNA elements, and phylogenetic analyses (*Atchley, Terhalle & Dress, 1999*). However, research on bHLH proteins in plants has been limited compared to that in animals, and therefore the exact organization of bHLH genes is unclear. Generally, the bHLH gene family in plants has been divided into 15–26 groups (*Toledo-Ortiz, Huq & Quail, 2003*), and sometimes up to 32 when atypical bHLH proteins are included (*Carretero-Paulet et al., 2010*). Based on sequence homology and phylogenetic relationships, 147 bHLH genes were identified in Arabidopsis and grouped into 21 subfamilies (*Toledo-Ortiz, Huq & Quail, 2003*), while 167 bHLH genes were detected in rice and formed 25 subfamilies (*Li et al., 2006*).

In plants, the bHLH gene family plays important roles in plant growth and development, metabolic regulation, and response to environmental changes. In Arabidopsis, the BR-Enhanced Expression and Phytochrome Interacting Factors (PIFs) had been reported to response to cold (*Kim et al., 2010*) and light (*Paik et al., 2017*). Paclobutrazol Resistance and Cryptochrome 2 Interacting BHLH are involved in flowering initiation and root initiation (*Ikeda et al., 2012*; *Liu et al., 2013*), with the MYCs acting as positive regulators of jasmonate biosynthesis (*Fernández-Calvo et al., 2011*; *Song et al., 2014a*). Furthermore, Glabra3, Enhancer of Glabra3 and Transparent Testa8 are involved in anthocyanin biosynthesis in Arabidopsis (*Petridis et al., 2016*; *Wen et al., 2018*). Recently, numerous bHLH TFs in plants have been suggested to respond to diverse abiotic stresses and improve plant stress tolerance, including to cold, salt and drought (*Sun, Wang & Sui, 2018*). Under salt stress, *bHLH39* increased the expression levels of stress-response genes in wheat, and thus improved the salt tolerance of *bHLH39*-overexpressing wheat plants (*Zhai et al., 2016*). A total of 18 bHLH genes in poplar have been found to respond to salt stress (*Zhao et al., 2018a*). In addition, *CgbHLH001* in *Chenopodium glaucum* can interact with *CgCDPK* in a signal transduction pathway under salt stress (*Wang et al., 2017b*). A study on grape validated the activity of *VabHLH1* and *VvbHLH1* as positive regulators under cold stress (*Xu et al., 2014*). *NtbHLH123* in *Nicotiana tabacum* is a transcriptional activator that can bind to the G-box/E-box motif in the *NtCBF* gene promoter, thus regulating the expression of stress-responsive genes and increasing the cold tolerance of *Nicotiana tabacum* (*Zhao et al., 2018c*). *FtbHLH2* in *Fagopyrum tataricum* is significantly induced under cold treatment and overexpression of *FtbHLH2* results in better cold/oxidative tolerance in transgenic Arabidopsis (*Yao et al., 2018*). Furthermore, *StbHLH45* in *Solanum tuberosum*
(*Wang et al., 2018*) acts as a positive regulatory factor under cold stress. The functions of bHLH gene family in plant under biotic and abiotic stresses are gaining increasing attention, and will be an important direction of future research (*Sun, Wang & Sui, 2018*).

Lotus is an economically important aquatic plant that has been widely used for food, medicinal, and ornamental purposes. As a basal eudicot plant with numerous monocot characteristics, lotus has been an important subject of evolutionary and taxonomic studies (*The Angiosperm Phylogeny Group, 2009*; *Ming et al., 2013*; *Zhao et al., 2018b*). Meanwhile, lotus plays a vital role in cultural and religious activities and is extensively distributed throughout Asia and Northern Australia. Soil and water salinity are some of the most serious environmental stresses affecting crop yields worldwide, so it is essential to improve the tolerance of plants to salinization of soil and water (*Munns & Tester, 2008*). In addition, low temperatures can severely impact the growth, yield and distribution area of plants (*Sanghera et al., 2011*). The dry matter content of lotus is minimal during colder seasons at high latitudes (*Bullard & Crawford, 2010*; *Halling, Topp & Doyle, 2010*), causing substantial economic losses. Hence, identification of salt and cold tolerance and response genes is expected to improve salt and cold resistance of lotus, which may increase the yield and expand the distribution of lotus to some extent. As noted above, the bHLH gene family responds to environmental stimuli, and thus structural and functional analyses of the *NnbHLH* gene family in lotus are necessary to improve stress tolerance in lotus. However, the bHLH gene family has not yet been comprehensively studied. Although the identification and structural properties of some bHLH family members were reported in 2014, systematic structural and functional analyses of the bHLH gene family in lotus have remained scarce (*Hudson & Hudson, 2014*). In this study, the bHLH gene family of lotus was systematically identified using bioinformatics methods. Then, thorough analyses of the gene sequences, gene structures and conserved motifs, phylogenetic relationships, gene ontology (GO) annotations, function prediction, and expression patterns were conducted. The results offer an effective framework for further functional characterization of the lotus bHLH gene family, and in particular their role under stress.

## MATERIALS AND METHODS

### Plant materials and treatments of the stress experiments

Seedlings (25 days old) of *Nelumbo nucifera* were grown in artificial climate chambers (Model:RXZ-500C, NingboJiangnan) with 16 h light and 8 h dark at 24 °C, following the conditions for lotus cultivation described in *Diao et al. (2014)*. For cold treatment, seedlings were placed in an Intellus Ultra Controller (Model:LT-36VLC8, Percival Scientific, Inc., Perry, IA, USA) with the temperature at 4 °C, a common condition for cold treatment, and the same photoperiod as described above. Leaves were collected before cold exposure (control), after 2, 4, 8, and 12 h of cold exposure for RNA extraction before expression analyses. For salt stress treatment, roots of the lotus seedlings were exposed to 50 mM NaCl (*Diao et al., 2014*), and the leaves were collected before treatment (control) and after 2, 4, 8, and 12 h of salt exposure. All of the samples were collected with

three biological replications and were immediately frozen in liquid nitrogen. The samples were stored at −80 °C for RNA isolation.

## Identification of the *NnbHLH* gene family

The lotus genome sequence and the gene annotation data were obtained from the lotus genome database (Lotus-DB: http://lotus-db.wbgcas.cn) (*Wang et al., 2015*). First, a Hidden Markov Model search (HMMsearch) was performed using HMMER software with the seed profile of the bHLH domain (PF00010) (*Cheng et al., 2018*; *Sun, Fan & Ling, 2015*), which was downloaded from the PFAM database (http://pfam.xfam.org/) (*Finn et al., 2014*). HMMER software was used to search for bHLH protein in the entire protein dataset with the *E*-value cut-off set to $10^{-5}$. Then, a Basic Local Alignment Search Tool protein (BLASTp) alignment against all lotus protein sequences was subjected to perform an extensive search for candidate bHLH genes using bHLH protein sequences from Arabidopsis and rice as queries. The protein sequences of Arabidopsis and rice were downloaded from PlantTFDB (http://planttfdb.cbi.pku.edu.cn) (*Jin et al., 2017*). The Simple Modular Architecture Research Tool (*Letunic, Doerks & Bork, 2015*) and Conserved Domains Search (*Marchler-Bauer et al., 2017*) were employed to detect the bHLH domain in candidate protein sequences.

## Chromosomal distribution, gene duplication analyses and calculation of synonymous (Ks) and non-synonymous (Ka) substitution rates

The distribution of each *NnbHLH* on megascaffolds was obtained based on the GFF3 file (*Wang et al., 2015*) and analyzed using Map Gene 2 Chromosome V2 (http://mg2c.iask.in/mg2c_v2.0/). Gene duplication analyses for lotus was conducted using the Multiple Collinearity Scan Toolkit (MCScanX) (*Wang et al., 2012*). To identify candidate homologous gene pairs ($E < 1e^{-5}$), BLASTp alignment was carried out across the whole lotus genome. The potential homologous gene pairs were identified, and then loaded into the program MCScanX with the default parameters to identify syntenic chains. MCScanX was used to further distinguish among whole-genome duplication (WGD)/segmental, dispersed proximal, and tandem duplication events in the *NnbHLH* gene family. Candidate homologous gene pairs identified in the same synteny block were applied to calculation of Ka and Ks values using DNAsp5 (*Librado & Rozas, 2009*).

## Sequence alignment and phylogenetic, gene structure and conserved motif analyses

Multiple domain alignments were performed using MEGA (vision 6.0) (*Tamure et al., 2013*) and loaded into Geneious to visualize. The phylogenetic tree was constructed using MEGA 6.0 with the neighbor joining (NJ) method and the following parameters: pairwise deletion and 1,000 bp replications. The phylogenetic tree was visualized by plotting it using the EvolView tool (http://www.evolgenius.info).

The intron-exon organization of *NnbHLHs* was visualized using the Gene Structure Display Server 2.0 (GSDS: http://gsds.cbi.pku.edu.cn/) (*Hu et al., 2014*). Conserved motifs

in *NnbHLHs* were identified using the MEME (http://meme-suite.org/index.html) (*Bailey et al., 2009*) server with the maximum number of motifs set to 20.

## GO annotation and analyses of cis-regulatory elements

Gene ontology analyses of *NnbHLHs* was performed using the Blast2GO program (*Conesa et al., 2005*), and the NCBI database was selected as the reference database. The sequences 1,500 bp upstream of the translation initiation codon ATG for each *NnbHLH* were selected for analyses of the promoters using a C script. Cis-regulatory elements for each promoter sequence were predicted through searching the PlantCARE database (*Lescot et al., 2002*) with the false discovery rate <0.1%.

## Expression profiles and qRT-PCR analyses of *NnbHLHs*

RNA-seq data of four lotus tissues (leaf, petiole, rhizome, root) were downloaded from Lotus-DB. Total RNA was isolated using an RNA extraction kit (Aidlab, Beijing, China) according to the manufacturer's instructions. 5X All-In-One RT MasterMix (abm) was used for reverse transcription. Specific primers for quantitative real-time PCR (qRT-PCR) were designed using Primer Premier 5 software (*Lalitha, 2000*). qRT-PCR was conducted using the LightCycler®/LightCycler®96 System Real Time PCR (Roche, Basel, Switzerland) with SYBR Premix Ex Taq II (TaKaRa, Kusatsu, Japan). The *actin* gene (GeneBank: XM_010252745) was used as an internal control for normalization of the expression levels of candidate *NnbHLHs* among different samples, following previous studies (*Cheng et al., 2013*; *Diao et al., 2014*; *Jin et al., 2016*), and expression levels were calculated using the delta–delta CT method (*Livak & Schmittgen, 2001*).

# RESULTS

## Identification, chromosomal distribution and physicochemical properties of *NnbHLHs*

Members of the bHLH family in lotus were identified in the lotus genome using two strategies, that is, HMM search and BLASTp search, as described in the Materials and Methods section. Then, to verify the sequences, all candidates were checked for the presence of a complete bHLH domain via CDD and SMART. In total, 115 sequences were confirmed as lotus bHLHs and named *NnbHLH1* to *NnbHLH115*. The encoded putative *NnbHLH* proteins were predicted to be 89 (*NnbHLH110*) to 779 (*NnbHLH61*) amino acids in length, with molecular weights ranging from 10.07 kDa (*NnbHLH110*) to 82.64 kDa (*NnbHLH61*) (Table S1). The grand average of hydropathicity values (GRAVY) of all candidate *NnbHLH* proteins were negative, ranging from −0.887 to −0.059, representing a hydrophilic characteristic. The predicted isoelectric points of *NnbHLH* proteins were 4.72 (*NnbHLH72*) to 10.62 (*NnbHLH110*). The predicted numbers of negatively charged residues (Asp and Glu) were 7 (*NnbHLH110*) to 104 (*NnbHLH1*), while the predicted number of positively charged residues (Arg and Lys) ranged from 18 (*NnbHLH110*) to 83 (*NnbHLH13*) (Table S1). Detailed information about these characteristics is listed in Table S1. The ratio of *NnbHLHs* in the *Nelumbo nucifera* genome was about 0.425%, similar to that in rice (0.44%) (*Li et al., 2006*) and poplar (0.40%) (*Carretero-Paulet et al., 2010*)
but lower than the ratios in Arabidopsis (0.59%) (*Toledo-Ortiz, Huq & Quail, 2003*) and *Brachypodium distachyon* (0.55%) (*Niu et al., 2017*). Based on the results of lotus genome annotation (*Wang et al., 2015*), the 115 predicted *NnbHLHs* were distributed unevenly, localized on 20 megascaffolds of lotus, with 168 megascaffolds in the lotus genome (Table S1). Among them, megascaffolds 1, 2, 3, 4, 5, 6, 7, 8, 10, 12, and 13 contained most *NnbHLHs*, while megascaffolds 9, 11, 15, 17, 24, 33, 43, 59, and 114 and scaffolds 554 and 634 only possessed 1–2 *NnbHLHs* each (Table S1).

## Multiple sequence alignment, prediction of DNA-binding and protein dimerization activity of *NnbHLHs*

To further clarify the structural characteristics of *NnbHLHs*, multiple sequence alignment analyses of the bHLH domain were carried out. As shown in Fig. 1, one basic region, one loop region and two helix regions were detected in all 115 *NnbHLH* proteins. And conserved amino acids in bHLH domains, with sequence identity more than 50%, were shaded in grey or black color (Fig. 1A). A sequence logo of the conserved amino acids in the *NnbHLH* domain was presented in Fig. 1B. The bHLH gene family in Arabidopsis, rice and *Brachypodium distachyon* all contain 17 conserved amino acids in the bHLH domain, which were also detected in the *NnbHLH* proteins (Fig. 1C; Table 1). The consensus ratios of the 17 conserved amino acids among the three species were also calculated (Table 1). Arg-16, Arg-17, Leu-27, and Leu-61 showed extremely high sequence identity (98%, 93%, 100%, and 94%, respectively) among the 115 *NnbHLH* proteins (Table 1). The HLH region may be essential for dimerization, particularly Leu-27 in helix 1 and Leu-61 in helix 2 (*Carretero-Paulet et al., 2010*). Thus, we speculated that *NnbHLH* proteins may also have the capacity to form protein complexes.

Based on the criteria developed by *Toledo-Ortiz, Huq & Quail (2003)*, bHLH proteins with more than five basic amino acid residues in the basic region were identified as DNA-binding proteins. *NnbHLH* proteins were divided into two major groups accordingly, including 101 DNA-binding proteins and 14 non-DNA-binding proteins. Then we subdivided the 101 DNA-binding *NnbHLH* proteins into two groups: 89 E-box-binding proteins (based on the presence of Glu-6 and Arg-9) and 12 non-E-box-binding proteins (without the simultaneous presence of Glu-6 and Arg-9), according to previous research (*Ellenberger et al., 1994*). The E-box-binding *NnbHLH* proteins were further subdivided into two groups, including 75 G-box-binding proteins and 14 non-G-box-binding proteins, based on the presence or absence of His/Lys-9, Glu-13, and Arg-17, which may be critical for binding to the G-box (CACGTG) (*Niu et al., 2017*; *Sun, Fan & Ling, 2015*).

## Gene structure and conserved motif analyses of *NnbHLHs*

A neighbor-joining phylogenetic tree was constructed using the alignment results of the *NnbHLH* proteins (Fig. 2A). Based on the bootstrap values, the *NnbHLH* proteins were divided into 19 subfamilies (Fig. 2A). Among those subfamilies, subfamily 16 was the largest, containing 16 proteins, while subfamilies 8 and 18 had only had 1 protein each.

Next, the exon/intron structures of *NnbHLHs* were analyzed using the GSDS 2.0 online tool. *NnbHLHs* in the same subfamily appeared to share similar numbers of exons and

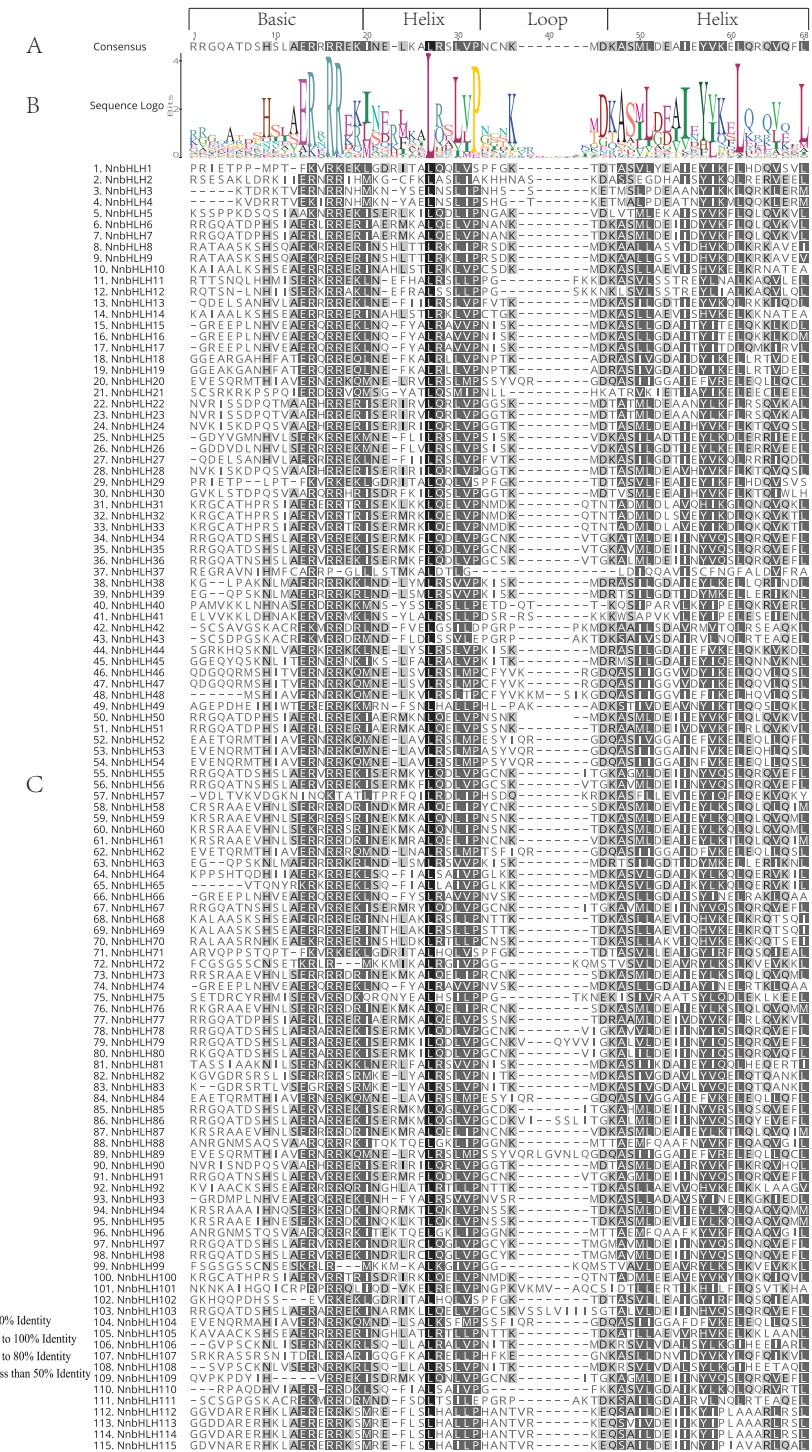

**Figure 1 Conserved amino acids and multiple sequence alignment schematic diagrams of the *NnbHLHs* bHLH domains.** (A) Conserved amino acids across *NnbHLH* domain. The amino acid with sequence identity more than 50% was labeled with gray or black shading (B) sequence logo of *NnbHLH* domains. The overall height of each stack represents the conservation of the sequence at that position. (C) Multiple sequence alignment of the *NnbHLH* doamins. Shading represents the degree of amino acids identity at each position, with the black shading indicating 100% sequence identity.

**Table 1 Consensus conserved amino acids in the bHLH domains between lotus, Arabidopsis, rice, and *Brachypodium distachyon*.**

| Position in the alignment | Consensus amino acids and their ratios within the lotus bHLH domain | Position in the alignment | Consensus amino acids and their ratios within the Arabidopsis bHLH domain *Toledo-Ortiz, Huq & Quail (2003)* | Position in the alignment | Consensus amino acids and their ratios within the rice bHLH domain *Li et al. (2006)* | Position in the alignment | Consensus amino acids and their ratios within the *Brachypodium distachyon* bHLH domain *Niu et al. (2017)* |
|---|---|---|---|---|---|---|---|
| 13 | E (82%) | 13 | E (76%) | 13 | E (68%) | 14 | E (74%) |
| 14 | R (85%) | 14 | R (74%) | 14 | R (67%) | 15 | R (73%) |
| 16 | R (98%) | 16 | R (91%) | 16 | R (90%) | 18 | R (88%) |
| 17 | R (93%) | 17 | R (86%) | 17 | R (84%) | 19 | R (88%) |
| 27 | L (100%) | 27 | L (100%) | 27 | L (99%) | 29 | L (99%) |
| 30 | L (81%) | 30 | L (65%) | 30 | L (68%) | 32 | L (69%) |
| 32 | P (92%) | 32 | P (88%) | 32 | P (82%) | 34 | P (56%) |
| 36 | K (70%) | | | 39 | K (76%) | 38 | K (64%) |
| 46 | D (66%) | 41 | D (64%) | 50 | D (71%) | 59 | D (68%) |
| 48 | A (77%) | 43 | A (73%) | 52 | A (74%) | 61 | A (69%) |
| 49 | S (63%) | 44 | S (53%) | 53 | S (57%) | 62 | S (51%) |
| 51 | L (75%) | 46 | L (76%) | 55 | L (86%) | 64 | L (84%) |
| 54 | A (51%) | 49 | A (60%) | 58 | A (57%) | 67 | A (58%) |
| 55 | I (72%) | 50 | I (63%) | 59 | I (64%) | 68 | I (55%) |
| 57 | Y (77%) | 52 | Y (78%) | 61 | Y (83%) | 70 | Y (77%) |
| 59 | K (61%) | 54 | K (55%) | 63 | K (62%) | 72 | K (53%) |
| 61 | L (94%) | 56 | L (93%) | 65 | L (96%) | 77 | L (99%) |

introns (Fig. 2B). Members of subfamilies 6, 7, 10, and 12 all contained the same number of exons and introns, while the majority of subfamilies 2, 9, 11, and 13 shared the same number of exons and introns. Meanwhile, the numbers of exons and introns were variable among subfamilies 1, 3, 15, and 19 (Fig. 2B).

To further reveal the specific regions of *NnbHLH* proteins, 20 conserved motifs that varied from 16 to 100 residues in length were detected using the MEME tool. All predicted motifs were identified only once in each *NnbHLH* protein sequence. *NnbHLH* proteins contained different numbers of conserved motifs, ranging from two to eight, and all *NnbHLH* proteins possessed motifs 1 and 2, representing the location of the bHLH domain (Fig. 2C). Meanwhile, *NnbHLH* proteins in the same subfamily always shared similar motif composition. For example, members of subfamily 9 all contained motifs 1, 2, 4, 10, and 11, while motifs 5, 17, and 19 were found exclusively in subfamilies 1, 12, and 14, respectively (Fig. 2C).

## Gene duplication analyses of *NnbHLHs*

Duplication events usually contribute to the expansion of gene families and genome evolution, and in particular, WGD/segmental and tandem duplication events have been important in the expansion of multigene families (*Kong et al., 2010*). In this study, almost all 115 detected *NnbHLHs* had experienced duplication events, except *NnbHLH21*. Because duplication usually contributes to the expansion of gene families, we further investigated the
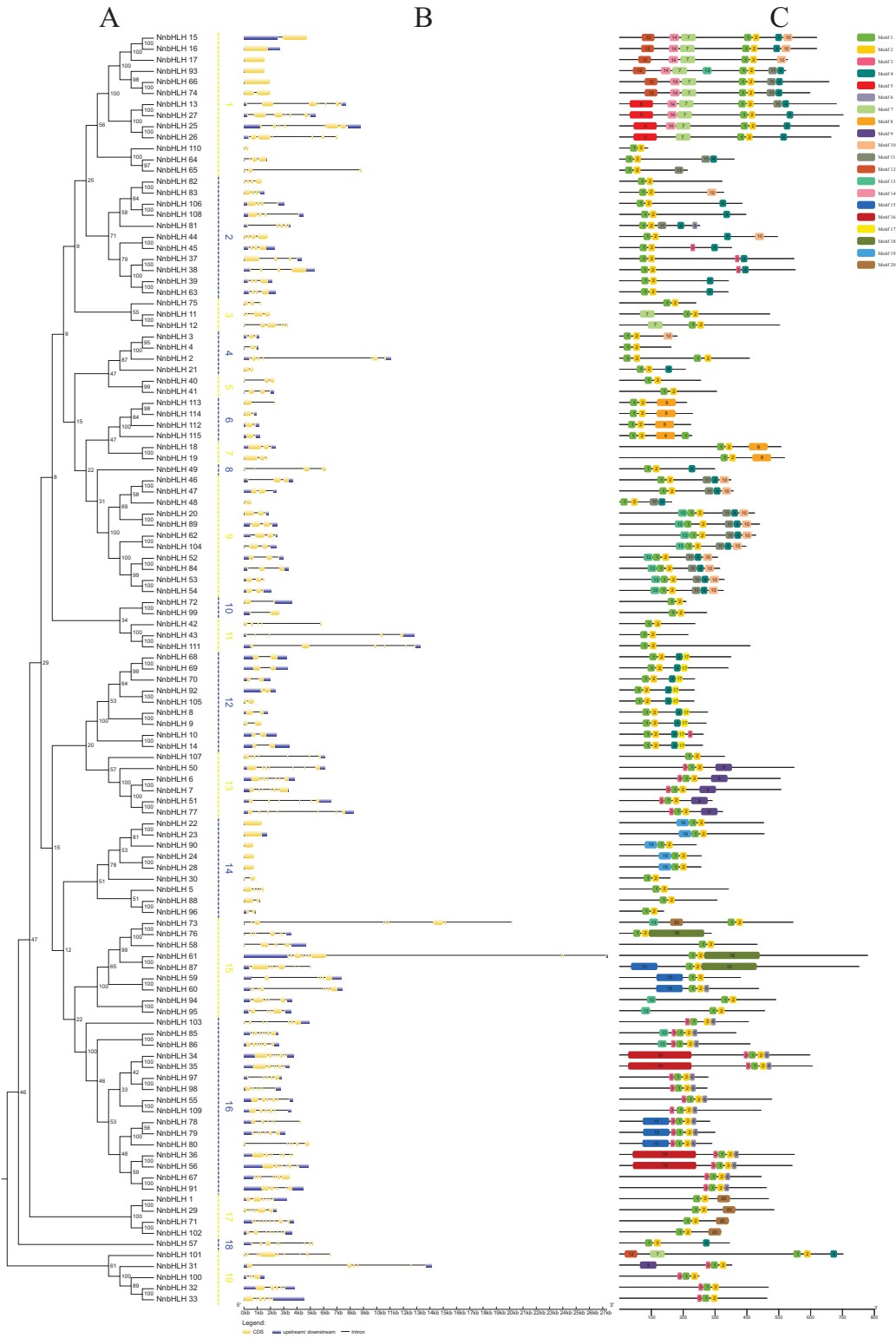

**Figure 2 Gene structures, phylogenetic relationships and conserved motifs analyses of the *NnbHLHs*.**
(A) Neighbor-joining phylogenetic tree of *NnbHLHs*. (B) Gene structure of *NnbHLHs*. Orange box represent exon, blue box represent UTR and the black line represent intron. The sizes of exons can be

Figure 2 (continued)
estimated by the scale at bottom. (C) Conserved motifs in the *NnbHLH* proteins. A total of 20 predicted
motifs are represented by different colored boxes and motif sizes can be estimated by the scale at the
bottom.                               

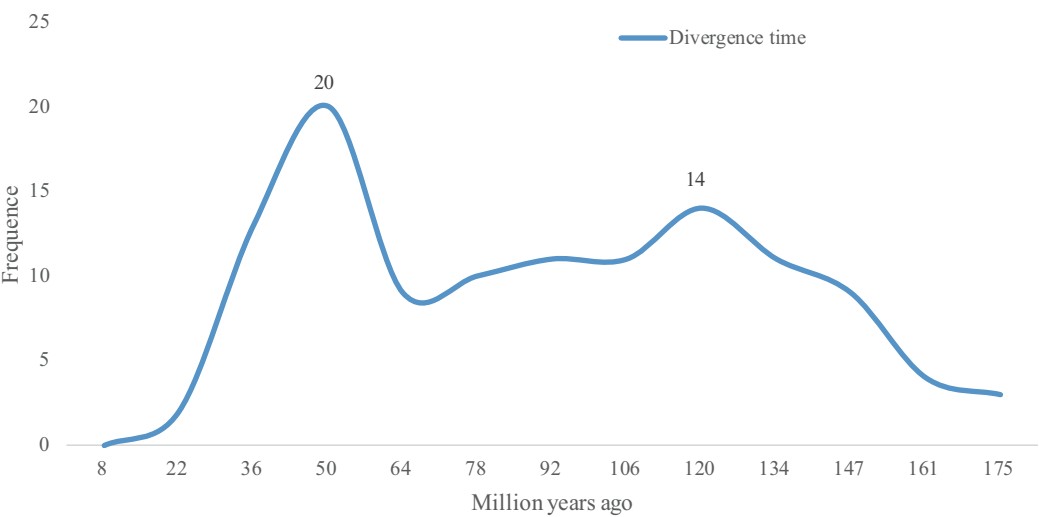

**Figure 3 Distribution of divergence time of WGD pairs of the *NnbHLHs*.** The *x*-axis represents
divergence time; *y*-axis represents the density of the distribution.

duplication patterns of each *NnbHLH*. In total, 80% (92/115) of the *NnbHLHs* were retained
from WGD/segmental duplication events (Table S2), with only 3.47% (4) from tandem,
13.9% (16) from dispersed, and 1.73% (2) from proximal duplication events.

The substitution rate ratio, Ka/Ks, is an effective criterion for the selective pressure during
gene duplication. Ka/Ks values less than one indicate negative selection, those equal to
one indicate neutral selection, and values greater than one indicate positive selection (*Yang,
2007*). Among the 118 pairs of WGD, the Ka/Ks ratios of 27 pairs were greater than one,
with those of *NnbHLH1* and *NnbHLH59*, *NnbHLH6* and *NnbHLH97*, *NnbHLH7* and
*NnbHLH76*, *NnbHLH26* and *NnbHLH93* being 1.808, 1.640, 1.510, and 1.679, respectively,
indicating that these duplication pairs have been strongly positively selected during their
evolutionary history (*Cheng et al., 2018*). The remaining *NnbHLH* genes, with Ka/Ks < 1, may
have undergone negative selection, suggesting that most bHLH genes have evolved slowly
(Table S2). For the two tandem duplication pairs, the Ka/Ks of the gene pair *NnbHLH112*
and *NnbHLH115* was 1.334, while that of *NnbHLH117* and *NnbHLH120* was 0.453.

As Ks values can be further be applied to trace the divergence time of duplication events
(*Lynch & Conery, 2000*), the divergence times of 118 WGD gene pairs were calculated.
As shown in Table S2, the divergence times of these 118 WGD pairs ranged from 8.39 to
167.35 Mya. The two peaks observed in Fig. 3 show that the lotus bHLH gene family has
undergone two larger duplication events, an ancient one that occurred ~120 Mya, and
a recent one that occurred ~50 Mya. The earliest duplication event in the *NnbHLH* family

appeared at 176 Mya. Since 147 Mya, bHLH family genes began to duplicate frequently, and the expansion of the gene family accelerated.

## GO annotation and cis-element analyses of *NnbHLH* proteins

To understand the specific functions of bHLH proteins, GO annotation of *NnbHLH* proteins was performed. As shown in Table S7, most *NnbHLH* proteins were annotated as being associated with protein dimerization activity, the development process, and response to stimulus. Within the cellular component, most genes were assigned to the nucleus (95/115). Very few genes had predicted distributions in organelles, such as the chloroplast envelope (8), mitochondrion (8), cytosol (4), and endosome (1). In addition, 19 *NnbHLH* proteins were predicted to be located on the plasmodesma (Table S7). In terms of molecular functions, almost all *NnbHLH* proteins were predicted to be involved in protein dimerization activity (113/115), with 83 proteins related to DNA binding TFs and 79 proteins associated with DNA-binding activity. These results are consistent with the DNA-binding analyses described in "Multiple sequence alignment, prediction of DNA-binding and protein dimerization activity of NnbHLHs." Within the biological process category, 102 *NnbHLHs* were predicted to be involved in development processes, including the development of the roots (52/115), carpels (43/115), floral organs (34/115), and petals (25/115). Moreover, 104 *NnbHLH* proteins were predicted to respond to stimuli, with 65 supposedly reacting to abiotic stresses. A total of 28 *NnbHLH* proteins could respond to low temperature, 35 were associated with responses to radiation (blue light, red light, or far-red light) and 13 were predicted to respond to salt stress. The number of *NnbHLH* proteins predicted to respond to abiotic stresses was high, suggesting that the *NnbHLH* gene family may play a vital role in lotus tolerance to abiotic stresses (Table S7).

Cis-element analyses were also conducted applying the 1,500 bp upstream sequence of *NnbHLHs* in the PlantCARE database. In the *NnbHLH* promoter region, G-box and Sp1 (response to light), MBS (response to drought), Skn-1_motif (required for endosperm expression), and ARE (essential for anaerobic induction) were the most common elements (Table S4). All *NnbHLH* family members except *NnbHLH 58, 86, 90, 100,* and *101* possessed at least one cis-element involved in stress responses (Table S4). The cis-regulatory elements present within *NnbHLH* promoters could be divided into three main categories. The first category was a ubiquitous class of plant light-responsive elements (such as G-box and Sp1), and the second category included plant growth- and development-responsive elements (such as Skn-1_motif and ARE) (Table S4). The last category included elements that respond to diverse stresses (such as TC-rich), including elements responding to biotic stresses (CGTCA-motif and ABRE) and abiotic stresses (HSE, MBS, and LTR). Elements in this category were widely distributed throughout the *NnbHLH* gene family (Table S4).

## Function prediction of *NnbHLHs* based on phylogenetic analyses

To date, the biological functions of most *NnbHLHs* have remained unclear. Meanwhile, in Arabidopsis and rice, the functions of many bHLH proteins have been characterized and verified. In this study, phylogenetic analyses allowed us to identify putative orthologous and paralogous bHLH genes in lotus, Arabidopsis and rice. In general, homologous genes
share similar structures and are clustered in the same clades, and these genes possess similar functions. To predict the gene functions of each *NnbHLH*, another NJ phylogenetic tree was constructed based on the protein of 115 *NnbHLHs*, 132 *OsbHLHs* of rice, and 160 *AtbHLHs* of Arabidopsis (*Li et al., 2006*; *Toledo-Ortiz, Huq & Quail, 2003*) (Fig. S1). In total, 24 subgroups were clustered, and the functions of *NnbHLHs* were predicted based on their homologs with verified functions in the same cluster (Table S5).

In summary, the majority of the members of subgroups 1a, 3, 4, 6, 7, 15a, and 19 may be able to enhance stress tolerance (*Kim et al., 2005*) and response to diverse abiotic and biotic stresses (*Sasaki-Sekimoto et al., 2014*; *Song et al., 2013*), including cold (*Kim et al., 2010*; *Deng et al., 2017*), salt (*Ahmad et al., 2015*; *Sakai et al., 2013*; *Jiang, Yang & Deyholos, 2009*), and drought (*Le Hir et al., 2017*). In subgroups 8 and 18, proteins may be related to Fe regulation, modulating the homeostasis of Fe content (*Kurt & Filiz, 2018*; *Wang et al., 2017a*). Members of subgroups 1a, 2, and 15a were predicted to regulate flower development of flower (*Sharma et al., 2016*; *Choi et al., 2018*; *Ito et al., 2012*), while those in subgroups 1b, 3, 4, and 17 may be involved in the development of various plant organs (*Karas et al., 2009*; *Le Hir et al., 2017*; *An et al., 2014*; *Lee, Jung & Park, 2017*; *Ohashi-Ito & Bergmann, 2006*; *Cui et al., 2016*; *Feng et al., 2017*). Subgroup seven members may regulate the flavonoid (*Rai et al., 2016*) and anthocyanin biosynthesis (*Wang et al., 2017a*). In subgroup 10, many proteins were predicted to be PIFs, which are related with photo-induced signal transduction and may optimize plant growth and development (*Paik et al., 2017*). Other members of subgroup 10 and subgroup 13 have also showed regulatory abilities (*Leivar et al., 2008*; *Zumajo-Cardona, Ambrose & Pabón-Mora, 2017*; *Cifuentes-Esquivel et al., 2014*). Members of subgroups 15b and 16 have been predicted to be involved in diverse processes (*Mai et al., 2015*; *Ferguson et al., 2017*; *Gaillochet et al., 2018*; *Zhu et al., 2016*). *NnbHLHs* in subgroup 21 should negatively control thermospermine biosynthesis and xylem differentiation (*Cai et al., 2016*; *Yamamoto & Takahashi, 2017*), while members of subgroup 24 may play an essential role in establishing vascular cells (*Ohashi-Ito, Matsukawa & Fukuda, 2013*). The detailed function prediction of *NnbHLHs* can be found in Table S5.

## Expression profiles of *NnbHLHs*

Transcriptional data of four different lotus tissues were employed to analyze the expression patterns of *NnbHLHs* (Table S6). In all tissues, *NnbHLHs* with FPKM > 1 were used for further analyses (*Zhuo et al., 2018*). Among 115 *NnbHLH* genes, 91 *NnbHLHs* were expressed in at least one of the tissues, while the expression levels of *NnbHLH 2, 12, 19, 21, 23, 30, 36, 40, 44, 45, 47, 48, 49, 61, 64, 65, 84, 88, 89, 90, 92, 96, 104,* and *110* were low, with FPKM < 1 in all four tissues (Table S6). Some *NnbHLH* genes exhibited tissue preferences, with expression levels in the preferred tissue that were more than two times higher than those in other tissues (*Sun, Fan & Ling, 2015*). Overall, *NnbHLH 20, 52, 84, 106, 108,* and *115* in the leaves; *NnbHLH 9, 24, 55, 97, 98,* and *109* in petioles, 14 genes (*NnbHLH 1, 4, 16, 29, 38, 50, 57, 59, 72, 74, 81, 93, 99,* and *102*) in rhizomes; and 22 genes (*NnbHLH 5, 11, 14, 26, 34, 39, 41, 46, 53, 54, 56, 60, 63, 67, 68, 69, 80, 82, 83, 91, 113,* and *114*) in roots were suggested to have tissue expression preferences (Table S6). In addition,

most members of subfamilies 4, 5, 6, 7, 8, 14, and 18 had no expression or low expression in the four tissues. The majority of the members of subfamilies 10, 11, and 19 exhibited high expression levels in all four tissues (Table S6).

Because stress-resistance is very important in lotus, many resistance-related *NnbHLHs* were predicted based on the structural and predictive analysis. In all, 15 *NnbHLHs* were selected for analyses of their expression under low temperatures and salt stress, and the specific primers used are listed in Table S3. The expression levels of the 12 *NnbHLHs* increased initially under 4 °C treatment (Fig. 4). Most were upregulated and reached their maximum values after 4 h of treatment, decreasing thereafter. The expression levels of *NnbHLH13* and *NnbHLH66* reached a maximum after just 2 h of treatment, while the expression levels of *NnbHLH16*, *NnbHLH38*, and *NnbHLH98* were highest after 8 h. We also found that *NnbHLH 15*, *17*, *37*, *38*, *55*, *66*, *70*, *81*, *93*, and *98* were significantly ($P < 0.01$) upregulated at all time points. Among these genes, the expression levels of *NnbHLH 15*, *17*, *55*, *66*, and *81* were significantly increased by 10-fold or more (Fig. 4); in particular, *NnbHLH17* and *NnbHLH81* were strongly induced under 4 °C treatment, with expression levels that increased by nearly 25-fold and 30-fold, respectively. Under NaCl treatment, eight *NnbHLHs* (*NnbHLH1*, *NnbHLH13*, *NnbHLH17*, *NnbHLH29*, *NnbHLH37*, *NnbHLH66*, *NnbHLH70*, and *NnbHLH81*) were upregulated initially and then decreased. Among these genes, only the expression level of *NnbHLH81* increased more than 10-fold compared to its untreated level. The other six *NnbHLHs* were downregulated. Although some *NnbHLHs* were significantly upregulated ($P < 0.01$) at one or two time points, none of them were upregulated at all time points (Fig. 4).

## DISCUSSION

### Comprehensive genome-wide detection of *NnbHLHs* in lotus

Numerous studies of the bHLH family in various species, such as Arabidopsis (*Toledo-Ortiz, Huq & Quail, 2003*), rice (*Li et al., 2006*), peanut (*Gao et al., 2017*), chinses cabbage (*Song et al., 2014b*), tomato (*Sun, Fan & Ling, 2015*), apple (*Yang et al., 2017*), and bamboo (*Cheng et al., 2018*), have been reported in recent years. In our study, 115 bHLH genes of lotus were identified, which is two fewer than in a previous study (*Hudson & Hudson, 2014*). This difference may be the result of a more restrictive selection in *NnbHLH* in our study, in which genes without complete bHLH domain were removed. These 115 genes were further classified into 19 subfamilies based on phylogenetic analyses (Fig. 2A). Multiple sequence alignment of the full-length *NnbHLH* protein sequences showed that all putative *NnbHLHs* contained the classic bHLH domain, and the number and ratio of the conserved amino acid were consistent with the Arabidopsis, rice, and *Brachypodium distachyon* (Table 1). Further structural analyses indicated that most of the *NnbHLHs* were associated with DNA-binding and homodimer formation activities. In addition, the conserved motif analyses showed that motifs representing the conserved domain were apparent in almost all the *NnbHLH* family members (Fig. 2C). Together, these results indicate that the 115 *NnbHLHs* all show characteristics of the bHLH family, which confirms the accuracy of the detection of bHLH gene family in lotus. Meanwhile, the

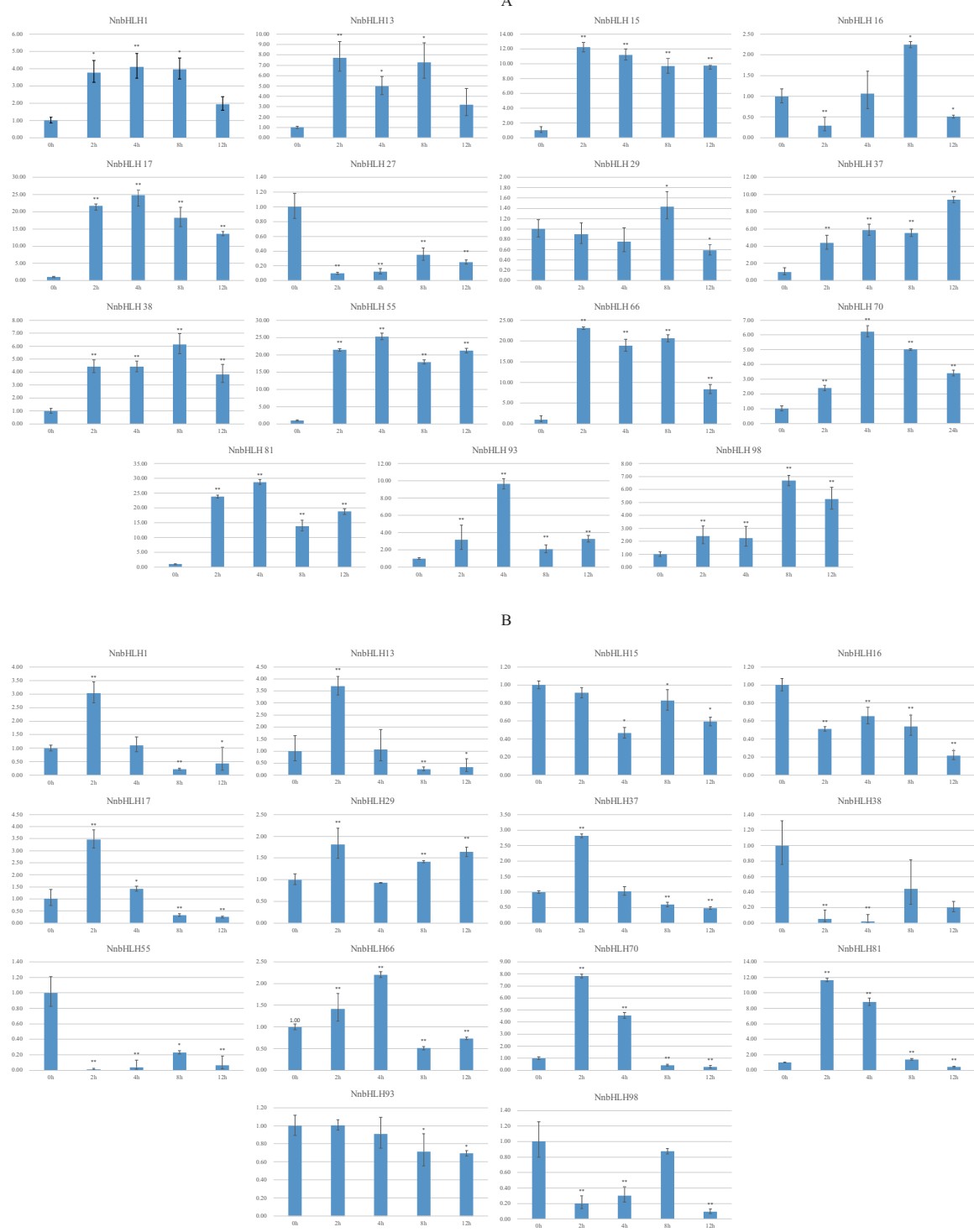

**Figure 4 Expression patterns of the *NnbHLH* candidate genes in lotus under cold and salt treatment by qRT-PCR.** (A) Expression patterns of *NnbHLH* candidate genes under cold stress treatment. (B) Expression patterns of *NnbHLH* candidate genes under salt stress treatment. *y*-axis: relative expression levels; *x*-axis: the time course of stress treatments; Error bar represent standard errors from three biological replicates. The asterisks indicate significant differences (*P* < 0.05) between treatment group and control group (0h) with two asterisks indicate extremely significant differences (*P* < 0.01).

expression profiles of *NnbHLHs* in four different tissues were analyzed, revealing that expression of many *NnbHLH* showed tissue preferences.

## *NnbHLHs* may play a special role in lotus evolution

The duplication pattern of a gene usually reveals how the gene was generated, how its function has evolved, and what roles it may play in plant growth and development (*Xia et al., 2017*). In our analyses, 80% (92/115) of *NnbHLHs* were duplicated through WGD (Table S2), similar to previous studies that have reported 48% in rice and 58.2% in *Brachypodium distachyon* (*Li et al., 2006*; *Niu et al., 2017*). Because the major duplication type in lotus was WGD, the bHLH gene family in lotus is presumed to have existed in ancient times (*Xia et al., 2017*). New genes produced through WGD usually increase the ability of plants to adapt to various growth conditions, suggesting that *NnbHLHs* have been indispensable to the growth and development of lotus (*Flagel & Wendel, 2009*; *Van De Peer, Maere & Meyer, 2009*). Meanwhile, most Ka/Ks values of WGD *NnbHLHs* gene pairs were less than 1 (Table S2), showing that the lotus bHLH gene family has evolved slowly (*Wang et al., 2013*).

Based on Fig. 3, the *NnbHLH* gene family underwent a large WGD duplication event in modern times (about 50 Mya). Interestingly, two extant species of Nelumbonaceae, the *Nelumbo nucifera* and *Nelumbo lutea*, reportedly diverged in modern times (*Xue et al., 2012*). Thus, the WGD duplication event observed in our study may be related to the divergence event of *Nelumbo nucifera* and *Nelumbo lutea*. Meanwhile, another peak appears around 120 Mya in Fig. 3, which suggests that the *NnbHLH* family experienced relatively frequent duplication events. Coincidentally, the Nelumbonaceae are closely related to Platanaceae phylogenetically, and their divergence time has been estimated to be around 75.2–122.8 Mya (*Xue et al., 2012*). A duplication event appeared around 120 Mya in our study, which may be related to the divergence of the Nelumbonaceae and Platanaceae. WGD events are related to the divergence of plant taxa and often appear to be accompanied by marked and sudden increases in species richness (*Van De Peer, Maere & Meyer, 2009*), and therefore the high frequency of WGD events in the *NnbHLH* gene family are likely to have been important to lotus evolution.

## Functional prediction of *NnbHLHs* in lotus

Research on the functional and structural genomics of Arabidopsis and rice has shown that bHLH TFs are involved in stress responses and plant development, including cold stress, heat stress, abscisic acid, jasmonic acid, and light response signaling pathways (*Ikeda et al., 2012*; *Kim et al., 2010*; *Liu et al., 2013*; *Paik et al., 2017*). Meanwhile, bHLH genes are involved in plant metabolite biosynthesis and trait development, including the formation of root hairs, anther development, and axillary meristem generation (*Fernández-Calvo et al., 2011*; *Song et al., 2014a*; *Wen et al., 2018*). However, little is known about the functions of the bHLH gene family in lotus. To better understand the second-largest gene family in plants, preliminary analyses of three aspects were conducted to reveal the functions of *NnbHLH* family genes in lotus for the first time.

Cis-element analyses showed that elements that can respond to diverse stresses (such as LTR, ABRE, TC-rich and HSE) were widely distributed in the *NnbHLH* gene family (Table S4). In addition, GO annotation revealed that most *NnbHLH* proteins were probably involved in the plant development process and responses to stimuli (Table S7). Furthermore, the functions of 69 *NnbHLHs* were predicted based on their known and verified homologs in Arabidopsis and rice (Table S5), which were mainly associated with development processes (root hair development, seed dormancy, fruit dehiscence, and flowering initiation) and stress responses (responses to low-temperature and salt stress and enhancement of stress tolerance) (Table S5). These analyses suggest that the bHLH gene family is also related to plant development, metabolic regulation, and the response to stress in lotus, in accordance with previous studies (*Li et al., 2006*; *Niu et al., 2017*; *Toledo-Ortiz, Huq & Quail, 2003*; *Zhang et al., 2015*). Next, we analyzed the candidate stress-response *NnbHLHs*, as improving stress tolerance in lotus is vital. Based on cis-element analyses, TC-rich cis-elements that may be involved in defense and stress responses were detected in the promoter regions of 58 *NnbHLH* genes (Table S4). Meanwhile, 38 *NnbHLHs* contained LTR box, which responds to cold stress and 44 members contained HSE box, which responds to heat stress (Table S4). The results of GO annotation suggested that 109 member genes can respond to stimuli (Table S7), with 13 and 28 *NnbHLHs* predicted to play roles in salt and cold stress, respectively, (Table S7). Phylogenetic analyses further suggested that 25 *NnbHLHs* may respond to stresses, including cold, salt, and drought, based on their homologs in Arabidopsis and rice (Table S5). Comprehensive analyses of these three datasets suggested that *NnbHLH 1*, *15*, *37*, *38*, *55*, *70*, *81*, *93*, and *98* and *NnbHLH13*, *15*, *16*, *17*, *27*, *29*, *66*, *70*, and *93* were likely to respond to cold and salt stress, respectively.

To validate the functional prediction of *NnbHLHs*, qRT-PCR analyses were conducted for 15 *NnbHLH* genes under cold (4 °C) and salt (50 mM NaCl) treatments. *NnbHLH 1*, *15*, *37*, *38*, *55*, *70*, *81*, *93*, and *98* were all significantly ($P < 0.01$) upregulated at all treatment time points, and were thus candidates of *NnbHLHs* that respond to cold stress. *NnbHLH 13*, *17*, *29*, *66*, and *70* were also upregulated under salt stress, as expected. We found that *NnbHLH 1*, *13*, *17*, *37*, *66*, *70*, and *81* were upregulated under both cold and salt stresses, in accordance with the results of GO annotations and cis-element and homolog analyses. Thus, *NnbHLH 1*, *13*, *17*, *37*, *66*, *70*, and *81* are the strongest candidates for lotus resistance genes due to their responsiveness to various stressors. The results of qRT-PCR analyses suggested that functional prediction of the *NnbHLH* gene family could provide valuable reference data for further functional research of this gene family.

## CONCLUSIONS

In summary, we conducted a genome-wide evaluation of the bHLH gene family in lotus. The structural characteristics of this gene family were thoroughly investigated. Functional prediction of the *NnbHLH* family was systematically conducted for the first time using three methods. We also analyzed the expression patterns of 15 candidate genes under cold and salt treatments at several time points based on functional prediction. Taken together,

the results and findings described in this study provide a strong basis for further investigation of the function and evolution of *NnbHLHs*. In addition, candidate genes for stress resistance in lotus were identified.

## ACKNOWLEDGEMENTS

We appreciate Dr. Tao Shi for his kindly help during data analyses.

### Funding

This research was funded by the National Natural Science Foundation of China (Project 31700619) and the Fundamental Research Funds for the Central Universities (No: 2662016QD022). The funders had no role in study design, data collection and analysis, decision to publish, or preparation of the manuscript.

### Grant Disclosures

The following grant information was disclosed by the authors:
National Natural Science Foundation of China: Project 31700619.
Fundamental Research Funds for the Central Universities: 2662016QD022.

### Competing Interests

The authors declare that they have no competing interests.

### Author Contributions

- Tian-Yu Mao performed the experiments, analyzed the data, prepared figures and/or tables, authored or reviewed drafts of the paper, approved the final draft.
- Yao-Yao Liu performed the experiments.
- Huan-Huan Zhu prepared figures and/or tables.
- Jie Zhang conceived and designed the experiments, contributed reagents/materials/analysis tools, authored or reviewed drafts of the paper, approved the final draft.
- Ju-Xiang Yang prepared figures and/or tables.
- Qiang Fu contributed reagents/materials/analysis tools.
- Nian Wang provides a server platform.
- Ze Wang prepared figures and/or tables.

### Data Availability

  The raw measurements are available in Dataset S1. The raw data shows the gene sequences of the *NnbHLHs* lotus family.

### Supplemental Information

Supplemental information for this article can be found online at http://dx.doi.org/10.7717/peerj.7153#supplemental-information.

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
