# Peer review of "Genome-wide analyses of the bHLH gene family reveals structural and functional characteristics in the aquatic plant Nelumbo nucifera"

_PeerJ, doi:10.7717/peerj.7153_

## Round 0.1 · original submission · Major Revisions

Dear Dr. Mao and colleagues:

Thanks for submitting your manuscript to PeerJ. I have now received two independent reviews of your work, and as you will see, the reviewers raised some concerns about the research. Despite this, these reviewers are optimistic about your work and the potential impact it will lend to research on bHLH gene family diversity. Thus, I encourage you to revise your manuscript accordingly, taking into account all of the concerns raised by both reviewers.

While the concerns of the reviewers are relatively minor, this is a major revision to ensure that the original reviewers have a chance to evaluate your responses to their concerns.

Please enlist the help of an English expert with your revision. Also, ensure that there is consistency across the analyses, tables and figures (legends particularly). Please follow standard gene and protein nomenclature. Make sure that all references suggested by the reviewers are included in your revision. Finally, avoid overstatements.

I look forward to seeing your revision, and thanks again for submitting your work to PeerJ.

Good luck with your revision,

-joe

·

Basic reporting

No comments

Experimental design

The reason for the authors to select the stress treatment should be described.

Validity of the findings

Acceptable

Additional comments

It is well known that bHLH gene family is a very large family containing over a hundred of genes in most of the sequenced plant species. Because of this, great attention has been paid on the functions of this gene family. However, it is still far away to fully understand their functions because of the diversification during gene evolution. In this study, the author conducted a comprehensive analysis on the members of this gene family in a newly sequenced plant lotus, which is very helpful. The whole manuscript and its analysis are acceptable for me. But I still have some minor concerns as listed below:

1, there is a mistake on reference citing about the lotus database. It should be Wang et al., 2015
2,The authors conducted cold and salt treatment without telling us the reason for them to choose these two, and why should be 4oC and 50 mM NaCl.
3, Based on figure 1B, it seems the sequences among different members are quite conserved, which seems to be conflict with figure 2C about the prediction of their conserved motifs.
4, The authors conducted gene duplication analysis, which must has also been performed in some other species, such as Arabidopsis and rice. Why did not they compare this results with those from Arabidopsis and rice, or even others?
5, Since the authors just analyzed the expression of these genes in 4 tissues, it is hard to determine the tissue-specific genes.
6, English writing may need to be polished further.

Reviewer 2 ·

Basic reporting

The manuscript by Mao and colleagues present a large identification and bioinformatic analysis of the bHLH gene family in sacred lotus. This approach will be useful for subsequent research programs, especially on stress resistance.
I however have some reservations about this manuscript, that to my opinion needs serious improvement before publication.

1- It should be corrected by a native English speaker.

2- In accordance with the international nomenclature, “NNbHLH” should be replaced by “NnbHLH”.

2- Figure and table legends are imprecise. For instance (not restricted to):
- Figure 1: For each panel, a first “title” sentence is missing. In the sentence “The amino acid with identify more than …”, “similarity” should be used instead of “identity”. In the last sentence, “similarity” should be replaced by “identity”.
- references in Tables 1 and S5 are not given
- two distinct tables are given under the name “Table S4”

3- Many mistakes in the author names in the “References” section, including (but not restricted to):
- Line 571: “Gabriela” is a surname, not a name. “Gabriela et al.” should be replaced by “Toledo-Ortiz et al.” throughout the manuscript.
- Line 583 : “Hir RL” should be replaced by “Le Hir R”.
- Lines 676, 682 and 688: many first names are indicated instead of family names.

4- Lines 189-190: sources of the ratios in other species?

5- Line 206-209: Ledent et al. (2002) does not report this.

6- Lines 239-247: the sentences are copy-pasted from Niu et al. (2017), cited in the manuscript. I feel some other sentences of the manuscript have been copy-pasted from other publications.

7- Lines 336-383 are a useless re-writing of Table S5

8- Line 397: “functional analysis” should be replaced by “predictive analysis”

9- Line 412: “the other 7” should be replaced by “the other 6”

10-Figure 5: subfamilies should be indicated. NnbHLH could be listed from 1 to 115, for easier reading.

Experimental design

11- The publication by Hudson and Hudson (2014), on bHLH family in sacred Lotus, should be mentioned in introduction rather than in discussion. The research question of the manuscript should be well explained, in view of this publication by Hudson and Hudson.

12- The growing conditions and treatments of the cold and salt stresses experiments are not well described.

13- In qPCR experiment, at least 3 housekeeping genes should be used.

14- Lines 387-395 : based on the figure, one cannot say that a gene is expressed specifically in a tissue, because the data are normalized by the average expression of bHLHs in each tissue.

15- Figure 6 (qPCR): what are the statistical tests? qPCR on control (non-treated) plants should be provided, to be sure that the variation in gene expression are not due to circadian regulation.

Validity of the findings

16- Figure 1: why are some protein sequences truncated at the N-term? In the analysis (Table 1), the authors do not take into account these truncated sequences to calculate the frequency of conserved AAs, which is not correct.

17- Figure 2: Subfamilies 16 and 17 should not be separated in such a way, because the most recent common ancestor of subfamily 16 proteins is also ancestor of subfamily 17 proteins.

18- Figure 2: bootstrap values should be indicated.

19- Lines 201-205: the list of given “conserved amino acids” are actually not the most conserved ones. They correspond to the conserved positions identified by Atchley and colleagues (1999) in animals, but do not correspond to the conserved ones in plants (as shown by Toledo-Ortiz and colleagues (2003) and by other publications). It would have been more accurate and more interesting to present the true list of conserved amino acids and to compare them to what has been published in Arabidopsis, rice, brachypodium,…

20- Lines 301-303: the category “DNA binding transcription factor” (85 proteins) is not listed.

21- Lines 314-316: the cis-element ARE (88 promoters) is not listed.

22- Line 332: the new phylogenic tree should be provided. Rice data are not presented.

23- Line 399, “the expression level of the 15 NNbHLH were all increased”: this is untrue, based on the figure.

24- Line 474, “Then GO annotation revealed that most NNbHLH proteins were involved in…”: this is an over-interpretation, as the GO annotation can only suggest this.

Additional comments

25- Line 192 : how many megascaffolds in total?

---

## Round 0.2 · Minor Revisions

Dear Dr. Mao and colleagues:

Thanks for revising your manuscript based on the concerns raised by the reviewers. I now believe that your manuscript is close for publication. However, one of our section editors has raised some concerns:

“The manuscript is well developed and is close to being ready to be accepted; however, there appears to be some problems with some of the files, the lack of being able to connect the GO annotations to the highlighted sequences, and that the sequences presented are not deposited within a sequence repository. As there are 15 candidate genes describes in four tissues and under stress conditions this data in prime for GO annotations in the biological, functional, and molecular categorization and is described as such; the connections need be placed somewhere, not in the histograms because they do not connect to the sequence data provided. Perhaps columns for GO terms can be placed in the supplemental file S6 and pointed to from the text of the manuscript. One of the supplemental files, S4 appeared very big and may need to be optimized for loading. On a machine with 16Gb of memory a statement was noted that said the column allotment was exceeded, and not all the data was loaded Maybe a gzipped CSV file would be better with proper headers. The sequence data is provided, but proper sequence annotation is missing; the name used should point to an already known reference sequence within one of the databases listed in the manuscript so that the author recognizes from where the sequence is derived ”

Please address these concerns as soon as possible so we may move towards accepting your work for publication.

Best,

-joe

·

Basic reporting

The authors have answerred all my concerns. It is now acceptable for me.

Experimental design

no comment

Validity of the findings

no comment

Reviewer 2 ·

Basic reporting

No comment.

Experimental design

No comment.

Validity of the findings

No comment.

Additional comments

I would like to acknowledge the work of the authors, that considerably improved the manuscript. To my opinion, it is now suitable for publication in PeerJ and would be very useful for the scientific community.

---

## Round 0.3 · accepted · Accept

Dear Dr. Mao and colleagues:

Thanks for submitting your manuscript to PeerJ. I have now received two independent reviews of your work, and as you will see, the reviewers raised some concerns about the research. Despite this, these reviewers are optimistic about your work and the potential impact it will lend to research on bHLH gene family diversity. Thus, I encourage you to revise your manuscript accordingly, taking into account all of the concerns raised by both
Thanks for again revising your manuscript. I now believe that your manuscript is suitable for publication. Congratulations! I look forward to seeing this work in print, and I anticipate it being an important resource for research communities studying bHLH gene family diversity. Thanks again for choosing PeerJ to publish such important work.

Best,

-joe

#